# Outdoor Characterization and Geometry-Aware Error Modelling of an RGB-D Stereo Camera for Safety-Related Obstacle Detection

**DOI:** 10.3390/s25247495

**Published:** 2025-12-09

**Authors:** Pierluigi Rossi, Elisa Cioccolo, Maurizio Cutini, Danilo Monarca, Daniele Puri, Davide Gattamelata, Leonardo Vita

**Affiliations:** 1Department of Agriculture and Forest Sciences (DAFNE), Tuscia University, 01100 Viterbo, Italy; pierluigi.rossi@unitus.it (P.R.); monarca@unitus.it (D.M.); 2CREA Consiglio per la Ricerca in Agricoltura e L’analisi Dell’economia Agraria (Research Centre for Engineering and Agro-Food Processing), Via Milano 43, 24047 Treviglio, Italy; maurizio.cutini@crea.gov.it; 3Department of Technological Innovations and Safety of Plants, Products and Anthropic Settlements, Italian National Institute for Insurance Against Accidents at Work (INAIL), Via Roberto Ferruzzi 38/40, 00143 Rome, Italy; d.puri@inail.it (D.P.); d.gattamelata@inail.it (D.G.); l.vita@inail.it (L.V.)

**Keywords:** agriculture, safety, RGB-D, depth cameras, obstacle detection

## Abstract

**Highlights:**

**Abstract:**

Stereo cameras, also known as depth cameras or RGB-D cameras, are increasingly employed in a large variety of machinery for obstacle detection purposes and navigation planning. This also represents an opportunity in agricultural machinery for safety purposes to detect the presence of workers on foot and avoid collisions. However, their outdoor performance at medium and long range under operational light conditions remains weakly quantified: the authors then fit a field protocol and a model to characterize the pipeline of stereo cameras, taking the Intel RealSense D455 as benchmark, across various distances from 4 m to 16 m in realistic farm settings. Tests have been conducted using a 1 square meter planar target in outdoor environments, under diverse illumination conditions and with the panel being located at 0°, 10°, 20° and 35° from the center of the camera’s field of view (FoV). Built-in presets were also adjusted during tests, to generate a total of 128 samples. The authors then fit disparity surfaces to predict and correct systematic bias as a function of distance and radial FoV position, allowing them to compute mean depth and estimate a model of systematic error that takes depth bias as a function of distance, light conditions and FoV position. The results showed that the model can predict depth errors achieving a good degree of precision in every tested scenario (RMSE: 0.46–0.64 m, MAE: 0.40–0.51 m), enabling the possibility of replication and benchmarking on other sensors and field contexts while supporting safety-critical perception systems in agriculture.

## 1. Introduction

Depth perception is one of the key abilities that allows both autonomous and semi-autonomous systems to move safely, plan their paths and understand their surroundings in most environments, playing a central role in what nowadays is defined as motion planning and obstacle avoidance. Over the past years, compact depth-sensing cameras—often referred to as RGB-D or stereo cameras—have become increasingly common thanks to their affordability, small size and ease of integration [1,2,3]: combining standard color red–green–blue (RGB) images with stereo triangulation, these devices provide real-time depth maps, making them useful across a large variety of applications, from mobile robots and logistics to driver-assistance systems and human–robot interaction [4,5].

Although these sensors are now widely used, their reliability in challenging environments is still questionable due to several aspects (weather, light, harsh terrain, etc.), and concerns arise when safety systems are solely based on such technology. In fact, most commercial products such as the Intel^®^ RealSense™ D400 series [6] are designed mainly for indoor or short-range robotic use [7,8]. When exposed to outdoor lighting, where brightness can severely change across a sunny day, cloudy conditions and dark environments, their performance naturally tends to drop. This often results in noisy depth data or systematic measurement bias [9,10]. Such limitations are the reason behind safety concerns, particularly in agricultural contexts, where accurate depth information is essential for avoiding collisions and detecting workers on foot [11].

Stereo vision estimates depth by identifying matching points in two or more images taken from slightly different positions and then triangulating their disparities: active stereo cameras expand on this principle by projecting infrared or structured-light patterns, which help find matches even in low-texture areas [12,13]. However, both active and passive systems are vulnerable to strong sunlight, reflections, shadows and vegetation, all of which can distort the captured patterns and reduce accuracy [14].

A good example is the Intel RealSense D400 series, which uses a global-shutter stereo pair, an optional infrared emitter and an internal processing pipeline for depth estimation. The camera’s Software Development Kit (SDK) includes several presets—such as default, high accuracy and high density—that balance precision, range and data density; however, their performance in outdoor conditions and outside of their standard range remains unpredictable and strongly context-dependent [15]. Prior research has shown, for example, that depth errors tend to increase toward the edges of the FoV [16], and that bright sunlight can amplify noise significantly [17]. Other studies confirm that ambient light and background texture can heavily influence the camera’s effective sensing range [18], also showing how agricultural environments pose some of the toughest challenges for vision-based perception [19,20,21]. Fields and orchards are visually complex, full of dense vegetation, fine textures and constantly changing lighting conditions [22]. Machinery operating in these environments must also contend with dust, shadows from plants or tools and oblique sunlight, all of which create strong and unpredictable contrast variations [23]. These factors make it difficult for stereo cameras to maintain stable and accurate depth estimates, especially at medium-to-long distances (4–15 m), where measurement precision naturally decreases, becoming a crucial factor for autonomous tractors or agricultural robots where the reliable detection of people and obstacles is a key safety aspect [24].

Depth cameras are a valid alternative to various sensing technologies that have been explored for agricultural safety: LiDAR sensors offer high accuracy at long ranges but are costly and provide lower resolution [25]. Ultrasonic sensors represent a low-cost solution but lack spatial precision. While purely visual systems based on RGB images can recognize humans or obstacles, they do not provide direct depth information [26]. Stereo cameras, therefore, present a promising solution, combining good accuracy, coverage and affordability when their errors can be properly characterized and corrected for real-world conditions.

Recent studies have started to address the need for the systematic characterization of stereo cameras in both lab and outdoor settings, given the limits of other technologies [27,28]. Moreover, while average error and standard deviation are common evaluation metrics, they do not fully describe how depth bias changes with distance or viewing angle. This is the reason why modeling such sources of bias is critical for predicting and correcting systematic errors, as well as for defining reliable confidence bounds that can inform safety-related decisions [29]. In addition, standards such as the International Organization for Standardization (ISO) 18497 [30] and ISO 25119 [31] define the performance requirements for systems that operate near humans, emphasizing the importance of measurable and validated sensor reliability; indeed RGB-D cameras, despite being highly attractive due to their affordable costs and high-resolution data, are limited by their performance in uncontrolled outdoor environments and in field deployments [32,33].

This work is built on previous research by presenting and validating a field-oriented protocol for testing stereo RGB-D cameras in realistic agricultural conditions. Using the Intel RealSense D455 as a case study, a series of controlled outdoor experiments were carried out to examine the effects of three key variables: (i) distance (4–16 m), (ii) field-of-view angle (0°, 10°, 20° and 35° off-center) and (iii) illumination time. A 1 m^2^ planar target was used for the tests and placed in vegetation to reflect the visual complexity of contexts such as agriculture and forestry. The goal is to map condition-dependent trade-offs that matter to safety functions, and to provide an error model that can be used to correct raw depth or, at minimum, to inflate safety margins when detection confidence drops.

To address the aforementioned gap between previous research and the use of RGB-D sensors in outdoor activities, an Intel RealSense D455 has been employed as a study platform to be tested in 128 outdoor trials that jointly vary three key factors, namely, distance, FoV and light conditions. The scope of the research is to estimate an interpretable, geometry-aware error surface that would also allow us to quantify the practical contribution of the built-in laser emitter projector of the camera at various ranges. The resulting model would provide valuable guidance for safety-related obstacle detection systems on activities that are highly affected by variability in environmental conditions, for instance, those involving agricultural machinery.

The protocol deliberately follows the camera’s built-in functions so that the resulting model stays anchored to the physics of stereo vision while remaining practical for field use, and it is meant to be applied to other cameras as well, since it is not just meant to identify the best calibration for the RGB-D sensor model that has been employed for the tests but to provide a model-agnostic methodology. Tests have therefore been organized to emphasize the effects of camera presets to obtain a compact and calibrated model that links depth bias to viewing geometry and illumination, which are elements that are directly linked to risk assessment models as well and system configuration settings as countermeasures.

## 2. Materials and Methods

Starting from the aforementioned state of the art on the use of RGB-D cameras for safety purposes, the authors selected an Intel RealSense D455 (Cupertino, CA, USA) as a reference, due to its widespread use in several applications and research studies [34]. Since, from a systems engineering perspective, understanding how perception uncertainty evolves with lighting and distance would allow adaptive safety margins, the authors intended to perform specific tests to evaluate the effects of environmental conditions on visibility and the performances of the sensor at different ranges and angles. As already stated in previous research, distance estimation is affected by an error which might severely limit their applications for safety purposes, given the chance of mistakenly assessing objects further than away than they effectively are. This is a major issue when the safety of vehicles depends on knowing the relative position of obstacles along their trajectory and when the braking distance is higher than the available space, generating the need for a model which is able to provide the error associated with a specific environmental condition and prevent dangerous optimistic estimations.

The research protocol developed for this research follows the camera’s native SDK processing pipeline and evaluates several built-in presets to gather several measurements in different configurations, using a disparity surface to identify and model systematic depth bias as a function of both range and angle of the surface in the Field of View (FoV). The purpose was to develop a model that could be used either to correct these biases directly or to estimate adaptive safety margins when confidence in sensor data is low, aiming to bridge the gap between controlled laboratory testing and the real-world outdoor deployment of stereo cameras. This will contribute to the development of standardized methods for evaluating depth cameras’ perception reliability, which is an essential step toward the safe and certified use of depth cameras in agricultural machinery.

### 2.1. Methodology for Depth Error Calibration

To do so, the authors designed a research protocol that takes into account both physics and concepts that lay behind the depth estimation methods, starting from how the distance of a pixel is estimated by the camera [35,36], as shown in Figure 1.

The ratio between the distance of the point from the origin (X) and the point’s distance (Z) equals the ratio of the position of the point in the left image (XL) and focal length (f). Equation (1) shows how XL can therefore be related to focal length and the point’s distance:(1)$$\frac{X}{Z}\;=\;\frac{X_{L}}{f}\;\;\;\;\;\;X_{L}=\;f\frac{X}{Z}$$

The same equation can be set up for the position of the point in the right image (XR), but of course the baseline’s length (B) must be subtracted from the X coordinate of the observed point. Equation (2) provides the same information regarding XL for XR as well.(2)$$\frac{X-B}{Z}=\frac{X_{R}}{f}\;\;\;\;\;\;\;X_{R}=f\frac{X-B}{Z}\;\;\;\;\;\;\;\;$$

Having obtained XL and XR, disparity (d) can be calculated as the difference among these two values. The result is shown in Equation (3):(3)$$d\;=\;X_{L}-X_{R}\;\;\;\;\;\;\;d=\;f\left(\;\frac{X}{Z}-\;\frac{X-B}{Z}\;\right)=f\frac{B}{Z}$$

From that information, distance (Z) can be calculated as follows in Equation (4):(4)$$Z=\frac{fB}{d}$$

This method is used to estimate depth in RGB-D cameras starting from a disparity calculated from two optical sensors of known focal length and baseline. To understand how these equations are affected by measurement errors, considerations must be found on derivative estimations on disparity in Equation (5).(5)$$\frac{\partial Z}{\partial d}=-\frac{fB}{d^{2}}\;\;\;\;\;\;\;\Delta Z\approx \left|\frac{\partial Z}{\partial d}\right|\Delta d=\frac{fB}{d^{2}}\Delta d=\frac{Z^{2}}{fB}\Delta d\;$$

Equation (5) provides the insights that are required to understand how distance error is generated and what would be required by a model to provide this kind of information. Given that the error is affected by the square of distance and of disparity, with the last being related to the angle at which the pixel is located in the frame, a model that considers these effects has been tested and has been defined, as shown in Equation (6), with the inclusion of both the assessed distance and the angle of the point plus several calibration weights and their relative interactions.(6)$$(Z,\;\alpha )=w_{0}+w_{1}Z+w_{2}Z^{2}+w_{3}tan\;\alpha +w_{4}({tan\;\alpha )\;}^{2}+w_{5\;}Z\;tan\;\alpha \;\;\;$$

This model aims to consider a series of parameters that vary according to the situation:an offset calibration value (w_0_) has been considered;scale errors (w_1_) related to the assessed distance;the quadratic growth of bias given by noisy measurements (w_2_Z^2^);a metric related to the FoV angle α, in a form (tan α) that aims to adhere to the X/Z ratio used in disparity, with its calibration weight (w_3_);the quadratic growth of bias given by extreme angle measurements, together with its specific weight (w_4_);overlapping effects of distances and angles, again with their weight (w_5_).

This choice on the mathematical model relies on the will to remain strongly linked to the physics of depth estimations and to leave some flexibility; a greater number of parameters could lead to better results based on test data, but also in overfitting the model to the camera type and field conditions. Moreover, since the model does not aim to interpret relative coefficient magnitudes nor apply scale-sensitive penalties, feature normalization is not required for unbiased estimation or prediction in this specific operational setting. The authors also assume zero-mean and mutually uncorrelated errors with respect to the weights and in relation to the values used as possible predictors (Z and tan α). Since the dataset consists of independent distance–angle trials, any time or spatial correlation tests would not be directly applicable, so focus has been given to an interpretable geometry-aware error model.

### 2.2. Experimental Design

Tests were run at the Experimental Farm of Tuscia University located in Viterbo, Italy. The Selected area was an environment that is common to many outdoor applications, namely, a plain lawn field of approximately 0.15 ha in size. Field layout, including the panel used for tests, is shown in Figure 2.

Having defined the field layout, points were selected to perform the tests. A set of four points were selected at distances of 4 m, 8 m, 12 m and 16 m. For each of these points, measurements were repeated in the following conditions:0 degrees, namely, with the camera right in front of the panel. In this condition, the panel will fall exactly in the center of the frame;10° angle, keeping the camera parallel to the panel, but with it being visible slightly on the right from the center of the frame;20° angle, with the same camera conditions of the previous scenarios but slightly more toward the edge of the frame;35° angle, which means that the panel is nearly on the edge of the frame, but is still fully visible at every distance.

The layout of the test points is shown in Figure 3.

For each of the 16 test points, a series of camera configurations were tested, mixing the high accuracy and high density mode with its laser emitter enabled or disabled. These are presets available in the Intel RealSense D4xx firmware, where high accuracy aims to provide better results for fewer pixels, while high density tends to provide the highest number of depth estimations in the frame; the laser emitter, instead, aims to support depth estimations in the short range to cover poor light conditions. The combinations that were tested, therefore, are as follows:High accuracy mode, laser emitter on;High accuracy mode, laser emitter off;High density mode, laser emitter on;High density mode, laser emitter off.

This experimental configuration leads to 64 diverse tests that had to be performed for each light condition scenario. To cover both good and poor light conditions, these tests were performed on a sunny day in September 2025 at 12:00 pm with approximately 64,000 lux of environmental light and on a cloudy day, still in September 2025, approximately at sunset (7:00 pm) with 14,000 lux of environmental light. Hence, these configurations resulted in 128 different tests (16 points, with 4 camera presets and in 2 different scenarios). The acquisition was performed by making the camera warm up for at least 10 min, while in the meantime verifying the proper inclination of the gray panel. As each test started, a .bag file was generated by running RGB and 2D depth captures for at least 5 s from the RealSense Viewer application for Windows 11 version 2.56.5. Frame resolution was set to 1280 × 720 pixels for both color and depth frames, each at 30 frames per second of acquisition rate.

### 2.3. Data Processing and Model Fitting

Data generated by .bag files was analyzed to identify depth parameters that could be useful for the calibration of the error on each test. The algorithm worked according to this flow of processing:Manual selection of the panel area in the frames. A python code reads all the bag files that are found in the folder in which it is located;The algorithm processes information such as number of pixels, hole rate of invalid depth values in the panel area, average depth, standard deviation of depth, number of frames and intrinsics;A comma separated value (csv) file is generated with all the metrics, along with a JSON (JavaScript Object Notation) file that stores data on the bounding boxes that have been defined on the panel area for each .bag file.

After having moved all the measurements from .bag files into a .csv file, the authors processed data to obtain features related to the error of distance compared with ground truth. Given the angle, laser emitter condition, high accuracy/high density settings, a ridge regression (or Tikhonov regularization) model, provided by the package scikit-learn of Python 3.11, was performed on the aforementioned Equation (6) to obtain weights for each scenario. The predicted bias provides corrected depth values according to Equation (6) and generated the following:A comparison of errors, for the same environmental and setting conditions, across distances for each test;Plots with approximate surfaces of areas sharing the same error according to distance and angle;A representation of the raw distance data with ideal condition (ground truth) and with the corrected values generated by the distance error model.

Lastly, data was saved regarding Root Mean Square Error (RMSE), Mean Squared Error (MAE) and weights of the regression model into another JSON file for subsequent usage in the field. The expected outcome of the tests is to verify error differences among test points and light conditions, with the additional feature of verifying the effect of the presence of the laser emitter to support depth readings, resulting in the possibility of analyzing each scenario separately (16 tests for 8 scenarios) or evaluating the possibility of merging the laser emitter on/off scenarios together (32 tests for 4 scenarios) to make a more robust regression model if the effects of the laser emitter in open fields are negligible.

## 3. Results

For each test, the authors had the possibility of identifying the test panel in the log file which presented both color and depth frames. Since data has been collected at different distances, the size of the panel area changed in pixel size across tests, leading to smaller detection boxes as the distance increased. The impact of such effects is also visible in far objects that fall beyond the maximum detection range of the camera, since they will most likely be identified as being located at the maximum available distance that the RGB-D sensor can provide, which is usually 64 m. An example of data visualization is shown in Figure 4.

Figure 4 already shows some of the classic effects of the two presets, high accuracy (HA) and high density (HD), since they provide a tradeoff between precision in depth measurements and the quantity of available measurements on each frame. For instance, the high density preset will provide a high quantity of data even on the ground around the target panel and of far elements, while the high precision mode will not provide much information on the ground surface while also reducing the number of measurements that will be available in the target area. Despite this not being an issue in close-range measurements, it becomes critical at long ranges, since the area of the target will be smaller and therefore will likely contain a limited number of valid values.

Results are therefore presented as depth estimations on each of the 128 tests, divided into scenarios, plus the error model estimation based on light conditions, distance and angle. Data for each scenario will also be analyzed afterwards to determine the effects of the laser emitter that is a built-in feature of the camera, which is a useful tool for indoor distance measurements but which might not be very effective in outdoor daylight conditions, given the distance of the targets and the presence of strong external lights, compared with indoor conditions.

### 3.1. Depth Distance Estimations

The outcome of field tests is a wide set of data divided into light conditions, panel angle position and distance from the camera. For each scenario, results are presented for the four presets. Tests on depth estimation in daylight with the laser off are shown in Table 1, where the data is shown with mean and median distances, standard deviation on the plane surface of the 2D panel, the hole rate and number of valid pixels, which also considers possible abnormal depth values which have been discarded.

Daylight test conditions without laser emitter support showed a progressively greater error as both the distances and angle of the panel position in the frame rise. In addition, the percentage of valid pixels (hole rate) changed significantly across the high accuracy and high density presets: despite the fact that lowered hole rate values were expected from the high density mode, the number of valid pixels in the plane area around the target in the frames was lower than expected.

Tests have also been run to compare outcomes with the laser emitter on and off, both in order to evaluate possible depth assessment improvements and to increase valid pixels. The results of these daylight tests that have been run with the laser on are shown in Table 2.

The same tests have been performed at sunset, both with the laser emitter off and on, as performed in daylight conditions. The results are shown in Table 3 and Table 4, where similar patterns observed for daylight have been confirmed in that environmental condition as well.

Distance estimations, as tested in sunset light conditions, started to be less precise as light is reduced; the absence of the laser emitter might be responsible for a decrease in precision at close distances but the assessments at far distances started to be significant as the angle of the target panel from the camera increased. It should be noted, however, that even the presence of the active laser emitter at maximum power did not change the result or improve depth readings, as was already observed in previous daylight tests with the laser emitter on. This is shown in the following table, which reports the last run of 32 tests in the given field positions from the target panel. The combination of the high accuracy preset and the reduced presence of external light played a key role in generating an even lower percentage of valid pixels in the target area across frames if compared with previous daylight tests.

For each test, further analyses were performed to investigate the influence of laser emitter conditions on depth readings, given that in outdoor environments the effect of this camera enhancement could have been limited. This part of the method is meant to verify if there are significant differences among the data and decide whether to perform error regression model estimations separately or not. The results are shown in Table 5, as mean results of the differences in the two corresponding readings for each test point.

No significant differences have been highlighted among the use of the laser emitter in the explored field tests; therefore, the upcoming analyses of the precision of the depth sensor at each condition will be explored as datasets of 32 readings (four distances at four angles each) in four scenarios, which are daylight with high accuracy and high density presets plus sunset scenario with high accuracy and high density presets.

### 3.2. Distance Error Comparison Among Scenarios

Test data has been compared across scenarios to verify how the error changed as the position of the panel went towards the edge of the frame. An important feature that needed to be assessed was the tendency of the error to vary as the distance of the panel from the camera increased. The results have hence been divided into each one of the four scenarios and depth estimation errors are shown in Figure 5.

In each scenario, the tendency of the error appears to linearly increase after 8 m, which is the limit of the expected functionality of the camera model to keep most of the errors below 0.5 or 1 m; beyond that threshold, errors rise and reach up to 3.5 m of error in the case of poor visibility or far distances, such as 16 m, which is twice the optimal range of the camera.

### 3.3. Ridge Regression Model for Distance Errors

The 32 tests for each of the four scenarios have been employed to run a ridge regression model to estimate the weights of Equation (6) that are necessary for the error prediction model. This aspect can show only slight differences in some weights, while it could have greater differences in others, such as those that refer to parameters that depend on the angle of the target. The result of the regression model on such weights for each scenario is therefore shown in Table 6, where RMSE and MAE are computed against the ground truth distances for each condition (daylight and sunset) and preset (HA and HD).

From the regression model, it is possible to see a high variability between daylight and sunset conditions, plus the expected change in the weights that affect the angle of the target. More specifically, despite RMSE and MAE remaining stable, the model shows a major impact of the W_3_ and W_4_ weights associated with the angle of the panel in the frame. Other parameters, such as W_5_, associated with the variability given by both distance and angle, remain stable within light conditions in both the high accuracy and high density presets.

### 3.4. Depth Error Estimation Based on Angle and Distance

The depth estimation model was employed to determine the depth error based on the angle and distance from the panel. The error can be represented by a surface on which it remains constant. Figure 6 shows the performance of the model in estimating the error in the four scenarios.

The iso-error surfaces in the previous figure show a consistent pattern across scenarios: the error, in fact, grows with range as expected but also intensifies as the target moves off-axis, producing steeper contours toward the edges of the FoV. Under daylight conditions, high density increases spatial coverage (i.e., more usable pixels), but of course it does not prevent angular bias; on the other hand, high accuracy yields slightly tighter surfaces in the central region while showing comparable off-axis behavior. In sunset light conditions, the angle-driven component becomes more prominent, with wider areas showing a higher degree of error, confirming that illumination modulates how strongly viewing geometry can generate an impact on depth stability. The plot also provides an intuitive representation of possible safety-critical zones in which the depth correction model can have a greater impact.

To better illustrate how this information can be translated into field use, it can be useful to compare raw depth readings with bias-corrected values from the regression model, as shown in Figure 7.

The compact ridge regression model captured the observed behavior with stable errors across scenarios. Terms linked to viewing angles emerged as especially influential under dimmer light, while interaction terms between range and angle helped explain the rapid growth of bias off-axis. The correction attenuates the systematic trend visible in the raw data, especially when overestimation or underestimation accumulates off-axis, bringing the observed distances closer to the nominal layout.

## 4. Discussion

For safety applications on agricultural machinery, the findings shown in this research translate into concrete design and integration guidance for both autonomous and manned vehicles, with the latter as an alert device or as a tool that provides extra visibility in blind spots. Detection zones that matter for braking and collision avoidance should be placed as close as possible to the optical axis, and system logic should assume greater uncertainty as targets move laterally or fall near the practical range limit. When illumination drops, planners should increase error margins or require other detection methods, since the angular component of the error becomes more prominent. Preset selection can be tuned to prioritize coverage when needed, but it will not replace geometry-aware handling of uncertainty. Embedding the proposed bias model in the decision stack allows distance-based rules to operate on corrected values or to adapt thresholds dynamically when conditions predict an increased risk of false negatives.

A consideration on the outcome of the trials is that the outer FoV sectors consistently exhibit larger errors, especially at medium-to-long ranges, suggesting that relying on peripheral areas for safety-critical operations should be avoided and, where possible, the camera should be configured so that the typical region of interest falls within the central area of FoV or at least should not exceed the 35° angle. For machines requiring wider coverage, dual cameras can be used to mutually cover edge regions, reducing the chances of abnormal distance assessments. On the software side, conservative gating can improve safety buffers, especially for peripheral areas of FoV at long range while keeping fairly good distance assessments in the central sector and at short range. Moreover, since daylight reduces the laser emitter effects on measurements, these recommendations can be considered to be applicable independently of the presence of any sort of laser projector on the camera.

The extremely low outdoor impact of the onboard emitter suggests that, in sunlight and against vegetation, passive stereo remains the dominant mechanism, with limited headroom for active assistance at the tested ranges. This supports a broader strategy that combines model-based correction with operational constraints and, where feasible, fusing complementary sensors to stabilize perception under challenging illumination. It can also be stated that, from a practical point of view, the developed model enables two complementary strategies, namely, applying the corrected distances from obstacles directly in motion planning (i) and alerting or inflating risk thresholds according to the degree of depth measurements’ uncertainty (ii). In both cases, the model can transform static readings driven by fixed presets into a geometry-aware and light-aware depth perception tool.

The limitations of this work are related to the use of a single device class, a controlled yet realistic background, and quasi-static scenes. Studies should, in future, repeat the same scientific protocol to broaden scene diversity, including articulated targets and dynamics to compare across stereo baselines and sensing technologies. Extending the protocol to moving platforms and publishing harmonized datasets would enable fair cross-device benchmarking and help translate model corrections into standardized safety envelopes usable across equipment and environments. Even within these bounds, the proposed approach offers a practical path to quantify, visualize and mitigate condition-dependent depth errors where it matters most: in the field, under real light, with actionable consequences for machine safety.

## 5. Conclusions

This study demonstrates that outdoor distance estimation with a stereo RGB-D camera is strongly conditioned by where an object sits in the FoV and how the scene is illuminated, with central regions yielding the most stable results and edge placement introducing systematic bias. This aspect is critical for the use of RGB-D cameras as part of safety devices or safety systems in working environments that are characterized by outdoor harsh weather conditions and by non-standardized obstacles that are typical, for instance, of agricultural contexts. The camera’s preset selection influences the balance between coverage and precision, yet does not remove the angular dependence of error, while the active projector offers little benefit at typical working ranges in outdoor applications. Instead, a compact and geometry-informed regression can capture these trends and enable bias-aware correction, turning raw depth into distances that align more faithfully with the physical scene, therefore allowing us to determine a better distance estimation based on environmental conditions.

Taken together, these findings argue for safety pipelines that explicitly manage field-of-view placement, monitor validity density and apply bias surfaces during risk evaluation rather than treating depth as uniformly reliable across the image. The proposed protocol and modeling workflow are readily transferable, providing a practical basis for acceptance testing, sensor placement and periodic recalibration in open-air agricultural settings. Embedding these steps into system integration can reduce optimistic distance readings in critical zones and support more dependable human–machine separation, with the purpose of creating a standardized and replicable practice across platforms and sites and allowing the possibility for future works to perform cross-model comparisons as well as testing additional challenging environmental scenarios such as night activities, operations under canopies or weather conditions such as rain and fog.

## Figures and Tables

**Figure 1 sensors-25-07495-f001:**
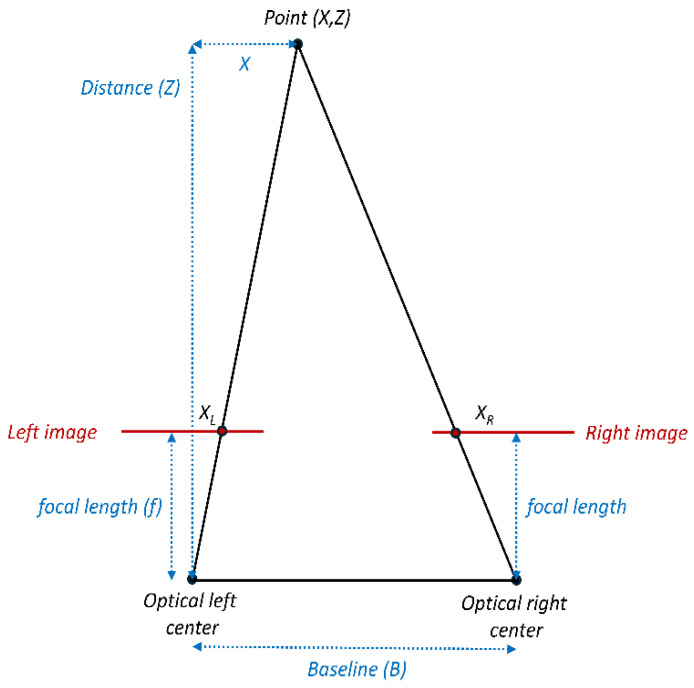
Disparity estimation method in depth cameras.

**Figure 2 sensors-25-07495-f002:**
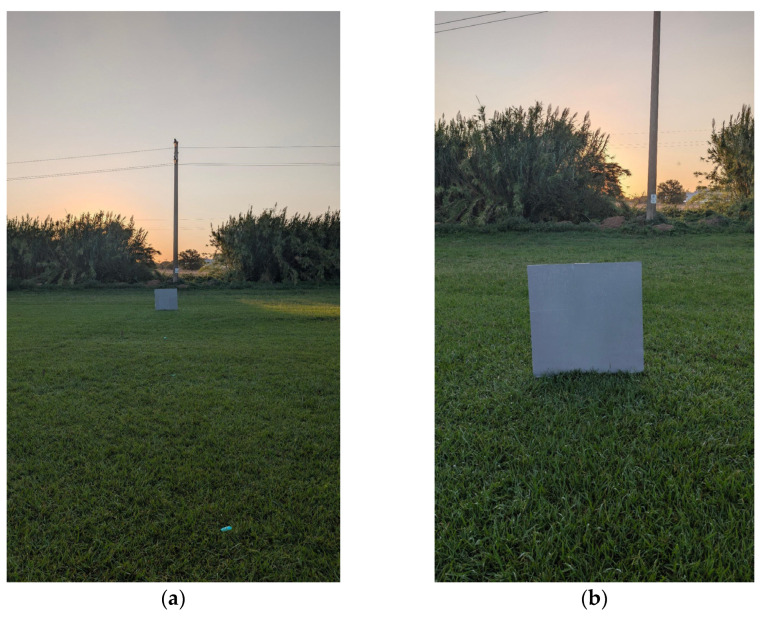
Field setup for detection tests. Placeholders were set on the grass as distance reference (**a**), while the gray panel was set parallel to the depth camera at each test point (**b**).

**Figure 3 sensors-25-07495-f003:**
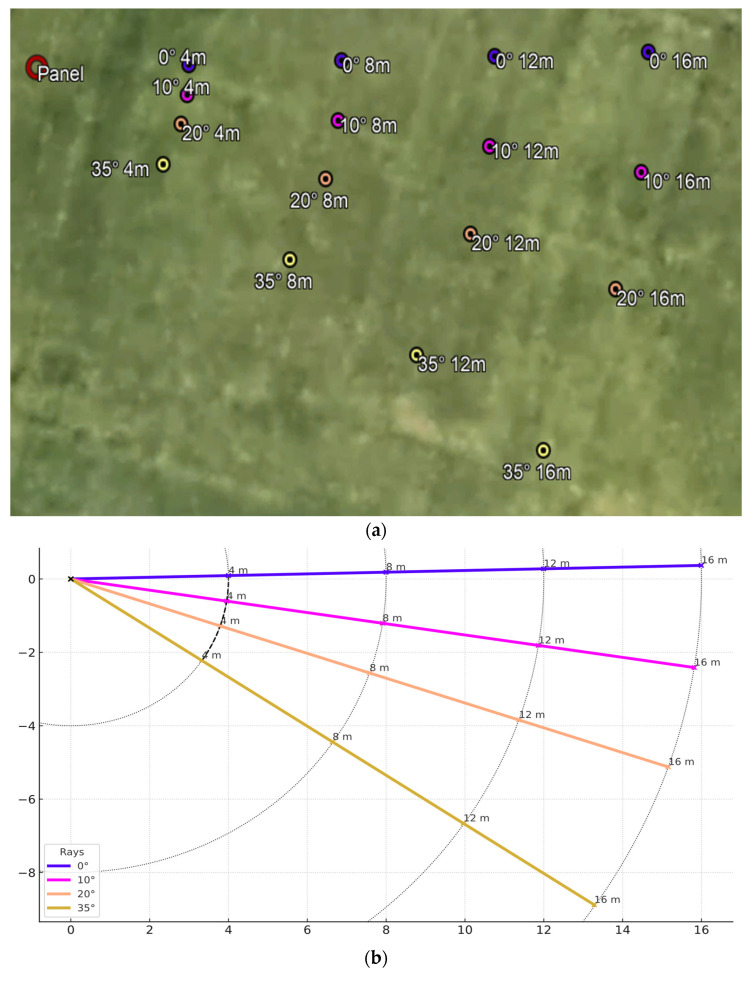
Test point positions on the map (**a**) and the criteria used to identify them (**b**).

**Figure 4 sensors-25-07495-f004:**
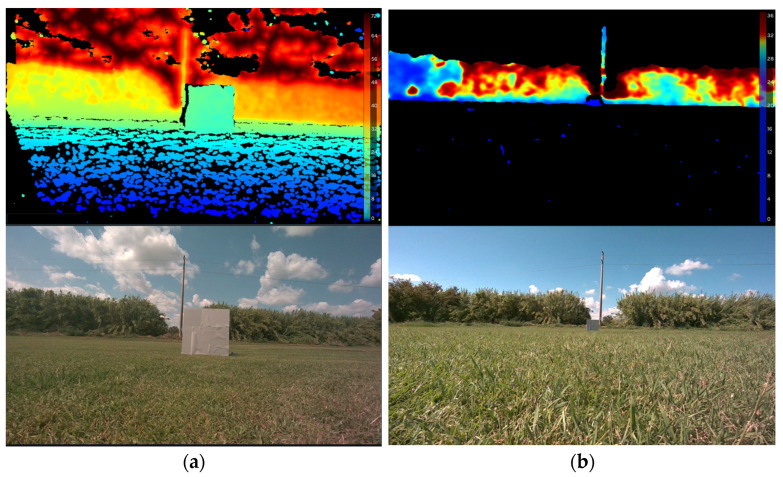
Example of test frames at close distance, high density preset (**a**) and far distance, high precision preset (**b**). Colors indicate depth from blue (close) to red (far), while black indicates no distance assessments.

**Figure 5 sensors-25-07495-f005:**
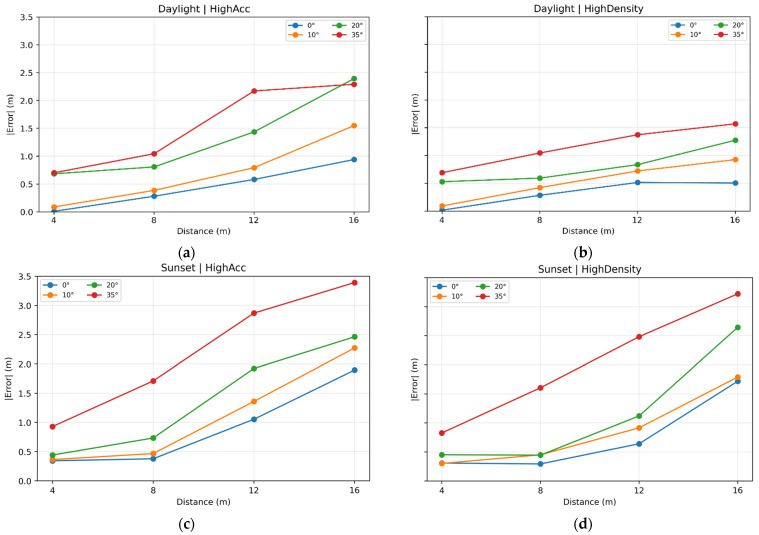
Error variability in each scenario: daylight and high accuracy preset (**a**), daylight and high density preset (**b**), sunset and high accuracy preset (**c**), sunset and high density preset (**d**), divided by positioning angle of the panel.

**Figure 6 sensors-25-07495-f006:**
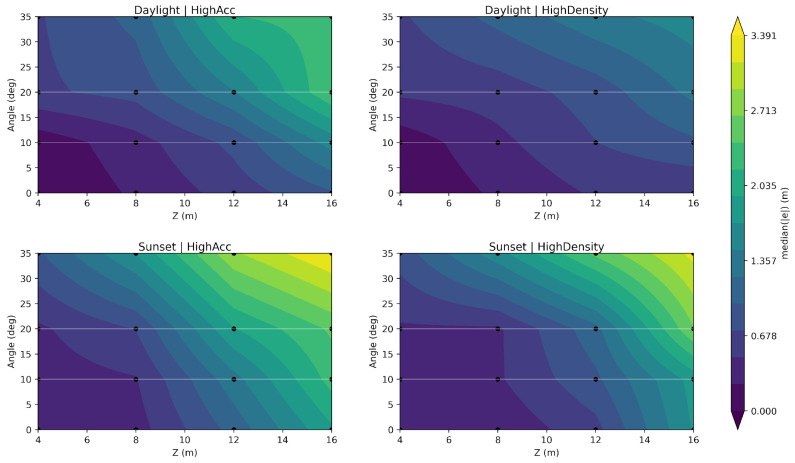
Depth error estimation according to the regression model for each scenario.

**Figure 7 sensors-25-07495-f007:**
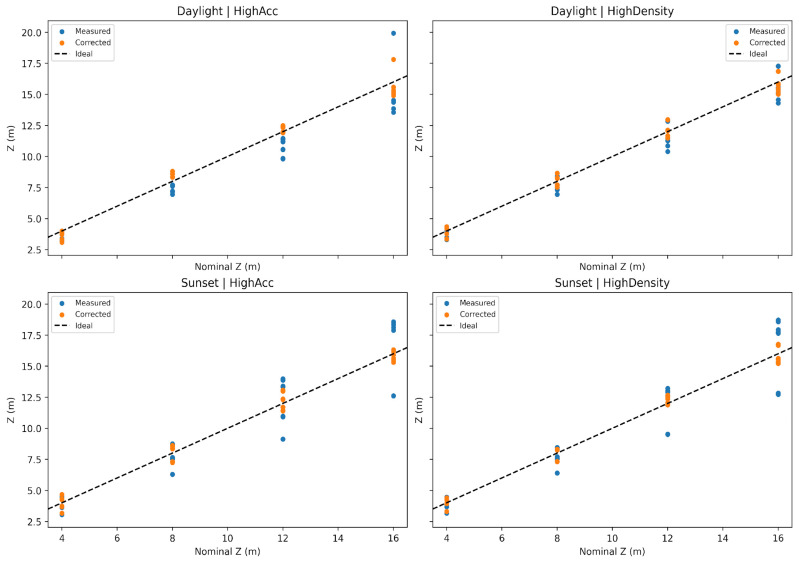
Comparison between raw depth estimation and model’s correction.

**Table 1 sensors-25-07495-t001:** Depth distance results in daylight conditions with laser emitter off. HA and HD stand for high accuracy and high density presets.

Distance	Position	Preset	Frames	Mean Distance	Median Distance	Plane Std Dev	Hole Rate	Valid Pixels
4 m	0°	HA	293	3.993	3.989	0.048	37.73%	62.27%
8 m	0°	HA	232	7.728	7.723	0.141	35.12%	64.88%
12 m	0°	HA	180	11.846	11.382	0.107	67.56%	32.44%
16 m	0°	HA	144	16.581	15.042	0.137	55.60%	44.40%
4 m	0°	HD	294	4.005	3.984	0.145	6.59%	93.41%
8 m	0°	HD	203	7.993	7.718	0.216	11.26%	88.74%
12 m	0°	HD	241	13.221	11.514	0.181	9.50%	90.50%
16 m	0°	HD	217	17.457	15.434	0.175	3.97%	96.03%
4 m	10°	HA	268	3.894	3.910	0.035	95.22%	4.78%
8 m	10°	HA	300	7.646	7.617	0.092	66.86%	33.14%
12 m	10°	HA	277	11.268	11.236	0.106	61.16%	38.84%
16 m	10°	HA	247	14.459	14.372	0.096	47.18%	52.82%
4 m	10°	HD	291	4.374	3.909	0.169	75.43%	24.57%
8 m	10°	HD	283	9.472	8.426	0.170	37.61%	62.39%
12 m	10°	HD	271	11.609	11.296	0.153	10.32%	89.68%
16 m	10°	HD	219	17.843	15.088	0.152	5.91%	94.09%
4 m	20°	HA	300	3.205	3.231	0.018	99.57%	0.43%
8 m	20°	HA	300	7.189	7.221	0.031	96.62%	3.38%
12 m	20°	HA	300	10.573	10.573	0.038	94.48%	5.52%
16 m	20°	HA	276	15.137	15.136	0.044	94.00%	6.00%
4 m	20°	HD	300	3.837	3.459	0.180	81.74%	18.26%
8 m	20°	HD	300	8.405	7.314	0.144	62.13%	37.87%
12 m	20°	HD	295	14.982	12.826	0.153	41.37%	58.63%
16 m	20°	HD	300	19.256	17.286	0.128	37.36%	62.64%
4 m	35°	HA	195	3.304	3.296	0.047	72.84%	27.16%
8 m	35°	HA	206	6.955	6.957	0.117	34.66%	65.34%
12 m	35°	HA	290	9.897	9.812	0.066	85.28%	14.72%
16 m	35°	HA	300	14.001	13.856	0.044	90.80%	9.20%
4 m	35°	HD	239	3.354	3.311	0.169	37.96%	62.04%
8 m	35°	HD	300	7.002	6.951	0.173	7.38%	92.62%
12 m	35°	HD	300	14.932	10.862	0.228	34.80%	65.20%
16 m	35°	HD	289	14.345	14.294	0.181	20.99%	79.01%

**Table 2 sensors-25-07495-t002:** Depth distance results in daylight conditions with laser emitter on.

Distance	Position	Preset	Frames	Mean Distance	Median Distance	Plane Std Dev	Hole Rate	Valid Pixels
4 m	0°	HA	300	3.994	3.990	0.049	38.05%	61.95%
8 m	0°	HA	184	7.717	7.715	0.144	36.01%	63.99%
12 m	0°	HA	195	11.524	11.454	0.136	39.91%	60.09%
16 m	0°	HA	206	15.818	15.078	0.140	54.81%	45.19%
4 m	0°	HD	249	4.052	3.983	0.242	7.41%	92.59%
8 m	0°	HD	191	7.900	7.714	0.212	8.60%	91.40%
12 m	0°	HD	169	13.130	11.457	0.194	7.89%	92.11%
16 m	0°	HD	175	17.700	15.555	0.169	4.11%	95.89%
4 m	10°	HA	234	3.897	3.914	0.039	95.28%	4.72%
8 m	10°	HA	263	7.650	7.611	0.100	60.93%	39.07%
12 m	10°	HA	212	11.165	11.175	0.100	57.42%	42.58%
16 m	10°	HA	249	14.774	14.528	0.108	53.11%	46.89%
4 m	10°	HD	300	4.550	3.908	0.174	75.61%	24.39%
8 m	10°	HD	276	8.654	8.419	0.172	44.14%	55.86%
12 m	10°	HD	271	11.519	11.261	0.155	9.84%	90.16%
16 m	10°	HD	212	17.850	15.059	0.148	6.19%	93.81%
4 m	20°	HA	300	3.324	3.402	0.023	99.02%	0.98%
8 m	20°	HA	272	7.151	7.165	0.028	97.43%	2.57%
12 m	20°	HA	270	10.517	10.554	0.037	92.45%	7.55%
16 m	20°	HA	262	20.547	19.924	0.078	94.55%	5.45%
4 m	20°	HD	300	3.896	3.484	0.177	82.26%	17.74%
8 m	20°	HD	300	8.822	7.499	0.149	62.61%	37.39%
12 m	20°	HD	300	13.596	12.846	0.137	40.50%	59.50%
16 m	20°	HD	281	19.235	17.265	0.144	33.75%	66.25%
4 m	35°	HA	187	3.308	3.300	0.046	74.05%	25.95%
8 m	35°	HA	300	6.957	6.954	0.112	37.24%	62.76%
12 m	35°	HA	266	9.887	9.845	0.101	70.29%	29.71%
16 m	35°	HA	300	13.958	13.561	0.050	89.93%	10.07%
4 m	35°	HD	208	3.353	3.308	0.171	40.76%	59.24%
8 m	35°	HD	271	6.983	6.957	0.170	6.86%	93.14%
12 m	35°	HD	300	13.397	10.390	0.207	36.50%	63.50%
16 m	35°	HD	300	14.873	14.565	0.187	19.34%	80.66%

**Table 3 sensors-25-07495-t003:** Depth distance results in sunset conditions with laser emitter off.

Distance	Position	Preset	Frames	Mean Distance	Median Distance	Plane Std Dev	Hole Rate	Valid Pixels
4 m	0°	HA	300	3.664	3.667	0.037	99.32%	0.68%
8 m	0°	HA	300	7.692	7.583	0.036	98.46%	1.54%
12 m	0°	HA	300	10.990	10.913	0.048	95.16%	4.84%
16 m	0°	HA	300	17.890	17.905	0.057	91.38%	8.62%
4 m	0°	HD	300	3.728	3.695	0.205	68.40%	31.60%
8 m	0°	HD	300	8.312	7.711	0.203	39.59%	60.41%
12 m	0°	HD	300	13.932	12.642	0.197	28.96%	71.04%
16 m	0°	HD	300	17.632	17.754	0.186	18.25%	81.75%
4 m	10°	HA	300	3.627	3.639	0.026	98.41%	1.59%
8 m	10°	HA	274	7.509	7.489	0.029	97.77%	2.23%
12 m	10°	HA	281	13.398	13.393	0.039	98.55%	1.45%
16 m	10°	HA	300	18.516	18.425	0.048	98.12%	1.88%
4 m	10°	HD	300	3.900	3.696	0.202	53.95%	46.05%
8 m	10°	HD	300	8.649	7.564	0.167	50.61%	49.39%
12 m	10°	HD	277	12.849	12.903	0.180	47.36%	52.64%
16 m	10°	HD	300	18.761	17.936	0.192	29.15%	70.85%
4 m	20°	HA	233	4.502	4.464	0.065	97.94%	2.06%
8 m	20°	HA	300	8.720	8.713	0.031	97.44%	2.56%
12 m	20°	HA	300	13.078	13.979	0.019	99.59%	0.41%
16 m	20°	HA	300	18.392	18.353	0.074	99.16%	0.84%
4 m	20°	HD	298	4.535	4.451	0.204	51.85%	48.15%
8 m	20°	HD	300	8.551	8.457	0.175	44.95%	55.05%
12 m	20°	HD	300	16.312	13.217	0.190	26.07%	73.94%
16 m	20°	HD	251	18.578	18.575	0.142	22.44%	77.56%
4 m	35°	HA	300	3.067	3.046	0.017	96.41%	3.59%
8 m	35°	HA	283	6.295	6.296	0.065	89.21%	10.79%
12 m	35°	HA	300	9.123	9.131	0.037	90.83%	9.17%
16 m	35°	HA	282	12.588	12.605	0.061	76.99%	23.01%
4 m	35°	HD	300	3.193	3.170	0.108	44.66%	55.34%
8 m	35°	HD	300	6.548	6.391	0.155	22.65%	77.35%
12 m	35°	HD	300	10.816	9.505	0.163	38.71%	61.29%
16 m	35°	HD	300	12.679	12.839	0.139	7.78%	92.22%

**Table 4 sensors-25-07495-t004:** Depth distance results in sunset conditions with laser emitter on.

Distance	Position	Preset	Frames	Mean Distance	Median Distance	Plane Std Dev	Hole Rate	Valid Pixels
4 m	0°	HA	300	3.644	3.647	0.035	99.26%	0.74%
8 m	0°	HA	300	7.674	7.661	0.051	97.96%	2.04%
12 m	0°	HA	300	11.038	10.979	0.047	94.92%	5.08%
16 m	0°	HA	300	17.945	17.887	0.052	95.22%	4.78%
4 m	0°	HD	300	3.710	3.686	0.210	68.47%	31.53%
8 m	0°	HD	300	8.691	7.701	0.213	40.24%	59.76%
12 m	0°	HD	300	13.721	12.641	0.192	28.38%	71.62%
16 m	0°	HD	300	17.588	17.684	0.180	16.49%	83.51%
4 m	10°	HA	298	3.633	3.632	0.037	97.55%	2.45%
8 m	10°	HA	300	7.707	7.577	0.035	97.00%	3.00%
12 m	10°	HA	300	13.348	13.326	0.030	98.81%	1.19%
16 m	10°	HA	300	18.252	18.120	0.054	95.82%	4.18%
4 m	10°	HD	300	3.897	3.698	0.205	54.77%	45.23%
8 m	10°	HD	300	8.696	7.533	0.168	52.84%	47.16%
12 m	10°	HD	291	12.812	12.930	0.161	45.28%	54.72%
16 m	10°	HD	300	18.843	17.645	0.182	29.03%	70.97%
4 m	20°	HA	300	4.539	4.419	0.033	98.11%	1.89%
8 m	20°	HA	300	8.965	8.754	0.036	97.44%	2.56%
12 m	20°	HA	284	13.895	13.866	0.035	99.46%	0.54%
16 m	20°	HA	223	18.469	18.575	0.097	99.11%	0.89%
4 m	20°	HD	267	4.515	4.454	0.204	52.25%	47.75%
8 m	20°	HD	300	8.435	8.434	0.172	44.56%	55.44%
12 m	20°	HD	280	15.576	13.025	0.182	27.50%	72.50%
16 m	20°	HD	300	18.986	18.715	0.156	22.07%	77.93%
4 m	35°	HA	300	3.098	3.095	0.015	97.52%	2.48%
8 m	35°	HA	300	6.282	6.283	0.063	90.69%	9.31%
12 m	35°	HA	300	9.119	9.132	0.030	91.73%	8.27%
16 m	35°	HA	279	12.613	12.612	0.062	77.38%	22.62%
4 m	35°	HD	300	3.192	3.174	0.109	46.96%	53.04%
8 m	35°	HD	296	6.698	6.397	0.157	23.54%	76.46%
12 m	35°	HD	265	10.843	9.528	0.150	40.99%	59.01%
16 m	35°	HD	296	12.690	12.718	0.149	8.80%	91.20%

**Table 5 sensors-25-07495-t005:** Mean difference in measurements among light conditions with laser emitter on and off.

Light Condition	Δ Distance (Mean)	Δ Distance (Median)	Δ Plane Std Dev	Δ Hole Rate	Δ Valid Pixels
Daylight	−0.060	−0.151	−0.007	0.013	−0.013
Sunset	−0.034	0.018	0.000	−0.003	0.003

**Table 6 sensors-25-07495-t006:** Weights and regression errors for each scenario.

Light	Preset	W_o_	W_1_	W_2_	W_3_	W_4_	W_5_	RMSE	MAE
Daylight	HA	2.543	−0.714	0.035	−0.837	−1.630	0.054	0.635	0.509
	HD	0.484	−0.217	0.010	2.606	−5.857	0.054	0.541	0.459
Sunset	HA	−0.493	−0.211	0.019	12.683	−17.073	−0.358	0.550	0.491
	HD	−0.278	−0.179	0.017	9.834	−13.237	−0.358	0.461	0.401

## Data Availability

Raw data (.bag files) from which datasets presented in this study have been derived are available on request due to their size.

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
