# Peer review of "Outdoor Characterization and Geometry-Aware Error Modelling of an RGB-D Stereo Camera for Safety-Related Obstacle Detection"

_sensors, 2025, doi:10.3390/s25247495_

Round 1

Reviewer 1 Report

Comments and Suggestions for Authors

This paper presents an evaluation of the Intel RealSense D455 depth sensor's performance at various distances and angles. The authors located an object to determine distances and angles, and then compared these measurements with those given by the sensor.
The authors characterize a model for estimating distance Z. That could be a better contribution if they can characterize other similar models of the Inter Real sense series, as done in the reference [12].

There is a methodological error when they adjust the model to the data. The data must be statistically normalized in order to compare relative parameter values and variations. Additionally, they must analyze the statistics of the residual of the model (whiteness) to determine if there are any non-modeled dynamics. 
Please add image examples of the capture data for large and small distances.
For agricultural applications, controlled illumination conditions during the night could be a possibility. It would be interesting to conduct experiments in low illumination conditions and at night, using artificial illumination. 

Observations:

Please explain in a better way the term "tan tan \alpha" in Eq. 6.

Improve the quality of Figure1.

Add x axis scale in Figure 4.

Reviewer 2 Report

Comments and Suggestions for Authors

The manuscript focuses on the agricultural machinery safety, addressing the issue of depth errors in RGBD stereo cameras in outdoor mid-range (4-16m) scenarios. By designing controlled experiments with three variables: "distance - field of view - lighting," it constructs a geometric perception error model and verifies its correction effectiveness. The research aligns with the practical needs of the agricultural automation field, demonstrating clear application value. The specific revisions to the paper are as follows:

  1. Specific Strengths: Error test of RGBD cameras was conducted in a realistic farm vegetation background (a 0.15-ha lawn with perennial ryegrass) instead of the traditional laboratory environment. The testing conditions are more consistent with the actual operating scenarios of agricultural machinery, making the research conclusions more guiding for the practical application of agricultural safety perception systems
  2. Introduction: Line 107-117, page 3, Strengthen the Connection between the Introduction and Subsequent Sections: The introduction only mentions the issue of "insufficient quantification of outdoor performance" but does not lay the groundwork for core research contents such as "three-variable interactive errors" and "ineffectiveness of the laser emitter". It is recommended to add 1–2 transitional sentences at the end of the introduction, e.g., "To address the above issues, this study used the Intel RealSense D455 as the research object, constructed 128 sets of outdoor experiments by controlling three variables (distance, FoV angle, and illumination), established a systematic error model based on disparity surface fitting, and verified the applicability of the laser emitter in medium-to-long range scenarios, providing a solution for the safety perception of agricultural machinery" to enhance the coherence of the "background-problem-solution" logic .
  3. Enhance the correlation between Discussion and Results: The application value of the "iso-error surface" in Figure 5 is not discussed in the Discussion section 4. It is recommended to add in the Discussion: "Figure 5 shows that the edge of the FoV (35° angle) is a high-error area, suggesting that the sensor installation of agricultural machinery should avoid setting detection blind spots at the edge of the FoV. A dual-camera cross layout can be used to cover this area and reduce collision risks", forming a closed loop of "results-application suggestions".
  4. Unify Terminology Expression: The terms "RGBD camera" and "RGB-D stereo camera" are used interchangeably throughout the manuscript (e.g., "RGB-D stereo camera" in title and "RGBD cameras" in text). RGBD=Red, Green, Blue, Depth. The inconsistent expressions of "field of view angle" and "FoV angle" should be unified as "field of view (FoV) angle" to avoid terminology confusion.
  5. In Equation (5) in page 4, “…b in the “square of Z divided by b” in “fb” should be an uppercase B instead of a lowercase b."
  6. In Equation (6), 𝜃 are not clearly defined. MAYBE it should be 𝛼. For every equation, it is necessary to add variable explanations after the equation.
  7. In Table 6, w0, w1, w2, w3, w4, w5 are not clear, scale error w1 is related to Equation (6). But the definition is not enough. It seems to miss something.
  8. The manuscript is a good topic: there are many RGB-D cameras on the market, of various types, but we know that depth recovery is an old issue and has been a research hotspot in the field of vision for many years, even today. In the conclusion section, the author's statements are somewhat ambiguous; how reliable are RGB-D depth cameras really? Readers will be very interested in this conclusion.

Round 2

Reviewer 1 Report

Comments and Suggestions for Authors

This second version of the papers considered my observations and were discussed in the Answer to the Reviewer document.

Please improve the quality of the figures. The PDF version I see has many artifacts and is low-quality.
